# Overall Survival of Hungarian Cancer Patients Diagnosed Between 2011 and 2019, Based on the Health Insurance Fund Database

**DOI:** 10.3390/cancers17101670

**Published:** 2025-05-15

**Authors:** Zoltán Kiss, Tamás László Berki, Anikó Maráz, Zsolt Horváth, Péter Nagy, Ibolya Fábián, Valéria Kovács, György Rokszin, György Surján, Zsófia Barcza, István Kenessey, András Wéber, István Wittmann, Gergő Attila Molnár, Tamás G. Szabó, Viktória Buga, Eugenia Karamousouli, Miklós Darida, Zsolt Abonyi-Tóth, Renáta Bertókné Tamás, Viktória Diána Fürtős, Krisztina Bogos, Judit Moldvay, Gabriella Gálffy, Lilla Tamási, Veronika Müller, Zoárd Tibor Krasznai, Gyula Ostoros, Zsolt Pápai-Székely, Gabriella Branyiczkiné Géczy, Lászlóné Hilbert, Csaba Polgár, Zoltán Vokó

**Affiliations:** 1MSD Pharma Hungary Ltd., 1095 Budapest, Hungary; 2Second Department of Medicine and Nephrology-Diabetes Centre, University of Pécs Medical School, 7624 Pécs, Hungary; 3Center for Health Technology Assessment, Semmelweis University, 1085 Budapest, Hungary; 4Department of Oncotherapy, University of Szeged, 6720 Szeged, Hungary; 5Department of Oncology, Bács-Kiskun County Teaching Hospital, 6000 Kecskemét, Hungary; 6Department of Molecular Immunology and Toxicology, National Tumor Biology Laboratory, National Institute of Oncology, 1122 Budapest, Hungary; 7Department of Anatomy and Histology, HUN-REN–UVMB Laboratory of Redox Biology Research Group, University of Veterinary Medicine, 1078 Budapest, Hungary; 8Chemistry Institute, University of Debrecen, 4032 Debrecen, Hungary; 9RxTarget Ltd., 5000 Szolnok, Hungary; 10Department of Biostatistics, University of Veterinary Medicine, 1078 Budapest, Hungary; 11Department of Deputy Chief Medical Officer II., National Public Health Center, 1097 Budapest, Hungary; 12Institute of Digital Health Sciences, Semmelweis University, 1085 Budapest, Hungary; 13Syntesia Medical Communications Ltd., 1065 Budapest, Hungary; 14Hungarian National Cancer Registry and National Tumor Biology Laboratory, National Institute of Oncology, 1122 Budapest, Hungary; 15Department of Pathology, Forensic and Insurance Medicine, Semmelweis University, 1085 Budapest, Hungary; 16Directorate of Institution, National Korányi Institute of Pulmonology, 1121 Budapest, Hungary; 171st Department of Pulmonology, National Korányi Institute of Pulmonology, 1121 Budapest, Hungary; 18Department of Pulmonology, Albert Szent-Györgyi Medical School, University of Szeged, 6720 Szeged, Hungary; 19Pulmonology Center of the Reformed Church in Hungary, Department of Pulmonology, 2045 Törökbálint, Hungary; 20Department of Pulmonology, Semmelweis University, 1085 Budapest, Hungary; 21Department of Obstetrics and Gynecology, Faculty of Medicine, University of Debrecen, 4032 Debrecen, Hungary; 22Fejér County Szent György, University Teaching Hospital, 8000 Székesfehérvár, Hungary; 23Demographic Statistics Department, Hungarian Central Statistical Office, 1024 Budapest, Hungary; 24National Tumor Biology Laboratory, National Institute of Oncology, 1122 Budapest, Hungary; 25Department of Oncology, Semmelweis University, 1085 Budapest, Hungary; 26Syreon Research Institute, 1145 Budapest, Hungary; 27Center for Pharmacology and Drug Research & Development, Semmelweis University, 1085 Budapest, Hungary

**Keywords:** cancer, survival, early cancer detection, Hungary, delayed diagnoses

## Abstract

This study from the HUN-CANCER EPI program analyzed cancer survival trends in Hungary between 2011 and 2019 using data from the National Health Insurance Fund. It is among the first to provide detailed overall survival information for Hungarian cancer patients, broken down by cancer type, sex, age group, and diagnostic period. The study found that some cancers—such as esophageal, pancreatic, liver, gallbladder, and lung cancer—had especially poor five-year survival rates, with many patients dying within just a few months of diagnosis. In contrast, testicular and thyroid cancers, Hodgkin’s lymphoma, melanoma, and breast cancer were linked to much better outcomes. These results underline the urgent need for earlier detection and focused treatment strategies. By identifying which cancers lead to the highest early mortality, the findings can help guide healthcare policy and improve support for patients, ultimately aiming to save more lives through targeted action.

## 1. Introduction

Although cancer is a major contributor to global mortality rates, survival prospects have significantly improved over the past few decades due to substantial advancements in treatment modalities and screening initiatives, particularly in developed countries [1]. However, certain types of cancer, including tumors of the lungs, pancreas, liver, and gallbladder, are still associated with poor prognosis and low 5-year survival rates, even in very-high-income countries [2,3,4,5]. Delayed diagnosis—often due to limited access to healthcare resources, socioeconomic disparities, and patient reluctance to seek medical attention—represents a major obstacle to improving cancer survival, as it prevents timely interventions and limits treatment options, leading to poor outcomes [6]. Addressing these barriers through improved screening programs, increased awareness, and better healthcare access is crucial for mitigating the impact of delayed cancer diagnosis on survival rates [7].

Overall survival (OS) is one of the key metrics in oncology (alongside net survival estimation), measuring the percentage of cancer patients who remain alive after a specified period from diagnosis. Together with net survival analyses, OS reflects advancements in treatments, early detection strategies, supportive care, and overall cancer management. It offers insights into treatment effectiveness and the impact of healthcare systems on patient outcomes [8]. Timely diagnosis remains a significant unmet need for various cancer types, as a relatively high proportion of patients are diagnosed at advanced stages and consequently have poor survival prospects [9,10,11]. On the other hand, a wide range of clinical studies has demonstrated improvements in OS rates for patients with colorectal cancer, breast cancer, and melanoma of the skin due to the advancements in treatment modalities [12,13,14,15]. Of note, previous studies have revealed significant sex disparities in cancer survival. Male patients tend to have worse outcomes compared to their female counterparts for lung, liver, colorectal, pancreatic, and gastroesophageal cancers, but they exhibit better OS rates for bladder cancer compared to women [16,17,18].

Hungary ranks among the countries with the highest cancer incidence and mortality rates in Europe. Therefore, conducting regular survival analyses of cancer patients using a well-established, comprehensive database is of utmost importance [19]. Nevertheless, detailed survival data—including sex- and age-specific information—are often lacking for certain cancer types. Although previous analyses from the population-based EUROCARE-5 project suggest relatively positive outcomes for Hungarian children with cancer, data on the broader spectrum remain incomplete [20]. Furthermore, these analyses generally do not cover the entire cancer population and lack information on specific cancer types.

### 1.1. Previous Epidemiological Research on the Cancer Burden in Hungary

The HUN-CANCER EPI study has provided detailed insights into age-standardized incidence and mortality rates for the most common cancers in Hungary, revealing a high cancer burden compared to European countries [21]. Among men, lung cancer had the highest incidence rate in 2018 (88.3/100,000 PYs), followed by colorectal cancer (82.2/100,000 PYs) and prostate cancer (62.3/100,000 PYs). By 2019, colorectal cancer surpassed lung cancer in incidence (87.2 vs. 79.3/100,000 PYs), reflecting a notable shift in the disease burden. Mortality rates for men remained highest for lung cancer (88.8/100,000 PYs), followed by colorectal cancer (46.5/100,000 PYs) and prostate cancer (20.9/100,000 PYs). Among women, breast cancer had the highest incidence rate (104.6/100,000 PYs), followed by lung cancer (47.7/100,000 PYs) and colorectal cancer (45.8/100,000 PYs). Mortality rates in women were highest for lung cancer (40.6/100,000 PYs), breast cancer (24.6/100,000 PYs), and colorectal cancer (22.6/100,000 PYs). Over the study period, overall trends showed a decline in age-standardized incidence and mortality rates for almost all cancers, with men experiencing a more pronounced decrease than women. These findings underscore the need for more detailed analyses of the Hungarian cancer population. While previously conducted population-based long-term survival analyses using data from the National Cancer Registry provided insights into changes in survival rates for common cancer types by stage over the past decades, showing improved survival for early-stage cancers [10], further data are required for more comprehensive analyses to effectively inform healthcare strategies.

### 1.2. Cancer Screening Programs in Hungary

Hungary has implemented organized cancer screening programs for several cancer types to reduce the disease burden through early detection. The cervical cancer screening program was officially launched in 2003 and offers Pap smear tests to women aged 25–65 every three years (participation rates are estimated at 30–50%) [22]. Organized breast cancer screening has been available since 2002, targeting women aged 45–65 every two years using mammography to detect early-stage breast cancers. Participation rates have been higher than those for cervical cancer screening but remain suboptimal at 45–55% over recent years [23]. There is no systematic national program for lung cancer screening in Hungary. Screening for lung diseases, including lung cancer, has historically been conducted through chest X-rays as part of the nationwide occupational health checks. Pilot studies have explored low-dose computed tomography (LDCT) [24], but no national program exists yet. Colorectal cancer screening was initiated as an organized nationwide program in November 2018, under coordination by the National Public Health Center (NPHC). The program, supported by European Union funding, offers fecal immunochemical tests (FIT) for individuals aged 50–70, with a follow-up colonoscopy for positive cases. Participation rates remain low, often below 30%, highlighting challenges in implementation and public engagement [25].

The primary aim of this analysis within the HUN-CANCER EPI study was to examine short- and long-term OS rates across the whole spectrum of prevalent cancers in Hungary during diagnostic periods spanning from 2011 to 2019. Furthermore, we aimed to identify cancer types with the highest early mortality rates within the Hungarian population by examining 6-month death rates and overall changes in survival. It is worth noting that parallel age-standardized net survival analyses from the HUN-CANCER EPI study are currently under publication; within this context, our OS calculations aimed to provide detailed insights into the exact survival metrics that can guide physicians and inform patients.

## 2. Materials and Methods

### 2.1. Data Sources

Our nationwide retrospective study used data from the Hungarian National Health Insurance Fund (NHIF) which covers the whole Hungarian population, providing detailed records on drug prescriptions, hospital admissions, outpatient consultations, and medical interventions, with diagnostic information classified according to the coding system established by the International Statistical Classification of Diseases and Related Health Problems, 10th Revision (ICD-10). Our study focused on individuals diagnosed with any type of cancer (excluding C44 and C77–80) between 1 January 2011 and 31 December 2019. Patient records from the NHIF database were identified based on social security numbers. The calculation of annual cancer incidence rates involved querying the NHIF database for individuals with at least two separate reimbursement records with cancer-related ICD-10 codes. For cervical cancer, two occurrences of a relevant ICD-10 code within 180 days were required, as detailed in a previous publication [21]. The patients who died within 60 days of the first appearance of a relevant ICD-10 code were also included in the analysis. The date of death was recorded by the Hungarian State Population Register Office, whose data are regularly updated in the NHIF database. The cancer cases diagnosed post-mortem were excluded from our analyses, as these diagnoses are not referred to the NHIF. When patients had multiple cancer-related ICD-10 codes, the code group with a higher frequency of occurrences was prioritized to determine the primary cancer type. This approach aimed to eliminate coding errors, such as the misclassification of metastatic lesions as primary tumors, an issue inherent to the reimbursement-oriented nature of the NHIF database. The cases involving secondary or multiple primary malignancies were excluded from subsequent analyses. For the patients with multiple cancer types, the “dominant” tumor type was determined based on ICD-10 codes with at least two occurrences. For international comparability, the patients were grouped based on different cancer sites as defined in Ferlay’s publications [19,26].

To accurately identify newly diagnosed cancer cases starting from 2011 and to exclude patients with prevalent cancers at the onset of the observation period, a prior observation period was set from 2009 to 2010. During this extraction, individuals with a history of cancer diagnosis codes before 2011 were excluded from further analysis. The date of diagnosis was defined as the first recorded occurrence of a relevant cancer-related ICD-10 code. The patients were subsequently monitored from the date of diagnosis until either the date of death or the conclusion of the study period on 30 September 2022. OS of the cancer patients was analyzed by age groups (0–18, 19–29, 30–39, 40–49, 50–59, 60–69, 70–79, ≥80), sex, and study year using data from the NHIF database.

### 2.2. Data Accuracy and Reliability

The accuracy and reliability of our data were ensured by the comprehensive nature of the NHIF database, which captures nearly 100% of the cancer cases treated within Hungary’s publicly financed healthcare system. This high level of coverage minimizes the risk of excluding major patient cohorts from the analysis. However, post-mortem cancer diagnoses, which are not included in the NHIF database, may lead to a potential underestimation of the overall cancer incidence. As these cases do not contribute to survival analyses, their exclusion does not affect the validity of our survival estimates.

To improve data reliability, we analyzed only one primary tumor per patient, determined based on the ICD-10 code group with the highest frequency of occurrence. This approach increases the likelihood of accurately capturing survival related to the patient’s dominant or primary cancer type, thereby avoiding errors such as misclassifying secondary or metastatic tumors as primary malignancies. While this methodology excludes secondary or multiple malignancies, it provides a focused assessment of survival outcomes for primary tumors.

Additionally, in our previous publication, we conducted sensitivity analyses on cancer-related interventions (e.g., surgery, radiotherapy, and systemic cancer treatment) to validate the cancer definition for each cancer type (Appendix A of [21]). We found that 81.51% of the total defined cancer cases involved at least one cancer-related intervention according to the aforementioned definition, which strengthens the validity of our approach.

### 2.3. Statistical Analysis

Survival assessment was initiated from the time of diagnosis, defined as the first occurrence of the dominant tumor-specific ICD-10 code for each patient. Censoring procedures were employed to account for patients who survived until the end of the follow-up period (from 1 January 2011 to 30 September 2022). OS rates were initially calculated using Kaplan–Meier estimates. Kaplan–Meier curves were generated for different cancer types, sex groups, and age categories. Differences in survival between groups were assessed for statistical significance using the log-rank test. To further evaluate survival differences across diagnostic periods, a Cox proportional hazards regression model was applied. This model analyzed the effect of diagnostic period on survival while adjusting for key covariates such as sex and age. Hazard ratios (HRs) with 95% confidence intervals (CIs) were estimated, and *p*-values were calculated to determine statistical significance.

The proportional hazards assumption underlying the Cox regression models was evaluated both statistically and graphically. Special attention was paid to the handling of censoring, with patients being right-censored at the end of the follow-up period (30 September 2022) if death had not occurred. The potential effects of informative censoring were considered minimal given the comprehensive national database coverage. Follow-up times were carefully calculated to guarantee consistent time origin definitions across the entire study population. Additionally, model adequacy was assessed by checking the influence of covariates and inspecting influential observations. Potential confounders such as age and sex were included as adjustment variables in the multivariate models. These steps were taken to ensure the robustness, validity, and reproducibility of the survival estimates presented.

This approach provided a more comprehensive understanding of survival changes between diagnostic periods. All statistical analyses were conducted using R version 4.1.2 (R Foundation for Statistical Computing, Vienna, Austria), employing the Survival package for Kaplan–Meier estimation and Cox regression modeling [27].

## 3. Results

### 3.1. Absolute Numbers

Between 2011 and 2019, a total of 528,808 patients were identified from the NHIF database with a diagnosis of any type of cancer (2011–2014: n = 238,195; 2015–2019: n = 290,613). Our analyses focused on the 2015–2019 period, during which we found the following patient numbers for the most common cancer types: colorectal cancer (C18–21): n = 45,792; lung cancer (C33–34): n = 45,548; breast cancer (C50): n = 37,941; prostate cancer (C61): n = 19,847; bladder cancer (C67) (n = 13,556); and melanoma of the skin (C43): n = 11,620, as shown in Appendix A. The highest patient numbers were found in the age cohort of 60–69 years (C00–97, excluding C44 and C77–80): 70,383 in 2011–2014 and 93,635 in 2015–2019. Detailed age-related data are shown in Appendix A.

### 3.2. Estimated Overall Survival by Different Types of Cancer

Figure 1 depicts 5-year survival estimates for patients diagnosed with various cancer types between 2015 and 2019. The malignancies associated with the lowest 5-year survival rates included esophageal cancer (C15): 7.0% (95% CI: 6.0–8.0%), pancreatic cancer (C25): 10.7% (95% CI: 10.1–11.3%), liver cancer (C22): 12.5% (95% CI: 11.5–13.5%), gallbladder cancer (C23–24): 13.9% (95% CI: 12.6–15.2%), and lung cancer (C33–34): 18.4% (95% CI: 18.0–18.8%). Conversely, higher survival rates were observed for testicular cancer (C62): 91.6% (95% CI: 90.6–92.7%), thyroid cancer (C73): 89.0% (95% CI: 87.9–90.0%), Hodgkin’s lymphoma (C81): 84.0% (95% CI: 81.7–86.2%), melanoma of the skin (C43): 78.6% (95% CI: 77.8–79.4%), and breast cancer (C50): 74.1% (95% CI: 73.6–74.6%).

Appendix A illustrates sex-related disparities in cancer survival rates. Our analysis revealed poorer survival outcomes for males compared to females across most cancer types diagnosed during the 2015–2019 period, except for gallbladder cancer and Hodgkin’s lymphoma. Specifically, 5-year survival rates among men and women were as follows: colorectal cancer = 43.3% (95% CI: 42.7–43.9%) vs. 47.2% (95% CI: 46.5–48.0%), lung cancer = 15.1% (95% CI: 14.6–15.5%) vs. 23.2% (95% CI: 22.5–23.8%), and melanoma = 73.2% (95% CI: 72.0–74.5%) vs. 83.6% (95% CI: 82.6–84.5%), respectively. Age-related differences in survival were analyzed for short-term (1- and 2-year) and long-term (3- and 5-year) intervals post-diagnosis across age groups (0–18, 19–29, 30–39, 40–49, 50–59, 60–69, 70–79, and 80 years and above). These findings, stratified by follow-up intervals (12, 24, 36, 48, and 60 months), are detailed in Appendix A.

Figure 2 shows changes in 5-year OS rates for the most common cancer types by age and sex. We found higher survival rates in females compared to males and in younger patients compared to elderly age cohorts. Furthermore, the patients diagnosed between 2017 and 2018 had a better survival probability compared to those diagnosed in 2011–2012. For instance, among the males diagnosed with colorectal cancer aged 60–69 years, the estimated 5-year survival increased from 45.1% (95% CI: 43.4–46.8%) during the 2011–2012 period to 48.1% (95% CI: 46.4–49.9%) during the 2017–2018 period. The corresponding survival rates in females in the same age group were 54.8% (95% CI: 52.7–56.9%) and 56.4% (95% CI: 54.2–58.6%), respectively. In general, survival rates were higher in younger age cohorts compared to older ones, both in men and women, for lung cancer, prostate cancer, and melanoma. However, among the female breast cancer patients, younger age cohorts had very similar survival rates to older cohorts (<70 years). Specifically, 5-year survival rates among the patients diagnosed in 2017–2018 were 87.6% (95% CI: 85.1–90.1%) and 80.9% (95% CI: 79.7–82.1%) in the patients aged 30–39 and 60–69 years, respectively.

### 3.3. Changes in 5-Year Overall Survival Between Diagnostic Periods (2017–2018 vs. 2011–2012)

The comparison of 5-year overall survival rates between the patients diagnosed in 2017–2018 and those diagnosed in 2011–2012 showed significant improvements across multiple cancer types, as illustrated in Figure 3. For colorectal cancer (C18–21), the hazard ratio was 0.76 (95% CI: 0.74–0.79; *p* < 0.001) in males and 0.78 (95% CI: 0.75–0.82; *p* < 0.001) in females, demonstrating significant improvements in survival for both sexes. Lung cancer (C33–34) in males showed a hazard ratio of 0.89 (95% CI: 0.87–0.91; *p* < 0.001), similar to that of females—0.89 (95% CI: 0.86–0.92; *p* < 0.001). Melanoma of the skin (C43) exhibited one of the largest improvements in males, with an HR of 0.57 (95% CI: 0.51–0.63; *p* < 0.001), and a similarly substantial improvement in females, with an HR of 0.62 (95% CI: 0.55–0.70; *p* < 0.001). Prostate cancer (C61), analyzed only in males, showed significant progress, with an HR of 0.65 (95% CI: 0.62–0.68; *p* < 0.001).

For breast cancer (C50) in females, the HR was 0.68 (95% CI: 0.66–0.71; *p* < 0.001), indicating notable survival improvement. Cervical cancer (C53) also demonstrated improved survival in females, with an HR of 0.82 (95% CI: 0.75–0.91; *p* < 0.001). Changes in the overall survival from 2017–2018 to 2011–2012 are detailed by sex and age in Appendix A.

### 3.4. Early Mortality

Figure 4 shows the percentage of patients who died within the first 2, 4, and 6 months after diagnosis for tumor types with the highest mortality rates during the 2015–2019 diagnostic period. Notably, almost one-third of the patients diagnosed with liver (33.2%), pancreatic (27.9%), and gallbladder cancer (29%) died within the first 2 months, with over 50% dying within 6 months. Similarly high early mortality rates were observed for esophageal (51.3%), stomach (42.9%), lung (41.7%), and brain cancer (39.2%) by the 6-month mark. Of note, for these highly fatal cancer types, the majority of deaths within the first 6 months occurred shortly after diagnosis, predominantly within the initial 2 months. Appendix A provides further insights into sex-related differences in early mortality rates for these tumor types.

Figure 5 (and Appendix A and Appendix A separately for males and females) presents data ranked by the absolute number of deaths among the individuals diagnosed with cancer in 2019. The following cancer types were associated with the highest absolute numbers of deaths within the first 6 months post-diagnosis: lung cancer (3617 deaths), colorectal cancer (1752 deaths), pancreatic cancer (1119 deaths), stomach cancer (612 deaths), and liver cancer (544 deaths). In contrast, the cancers with the lowest number of deaths during this period included cervical cancer (102 deaths), skin melanoma (68 deaths), thyroid cancer (37 deaths), Hodgkin’s lymphoma (20 deaths), and testicular cancer (12 deaths).

## 4. Discussion

The nationwide HUN-CANCER EPI study marks a pivotal achievement in providing comprehensive survival data for the entire cancer population diagnosed in Hungary between 2011 and 2019. As the first analysis of its kind in the Central Eastern European region, it delivers detailed survival estimates for both short-term (1- and 2-year) and long-term (3- and 5-year) outcomes across various demographic factors, including age, sex, and different diagnostic periods, and a broad spectrum of cancer types.

The main findings of our study can be summarized as follows:During the 2015–2019 diagnostic period, the poorest 5-year OS rates were observed for cancers of the esophagus, pancreas, liver, gallbladder, and lungs. Conversely, tumor types such as testicular cancer, thyroid cancer, Hodgkin’s lymphoma, melanoma, and breast cancer exhibited better survival outcomes. Additionally, an improvement in the overall survival was observed across nearly all cancer types when comparing the 2017–2018 and 2011–2012 diagnostic periods.We found sex-related disparities in OS, with males generally exhibiting worse survival compared to females across most cancer types, except for gallbladder cancer and Hodgkin’s lymphoma.In general, survival rates were higher in younger age cohorts compared to older ones both in men and women for lung cancer, prostate cancer, and melanoma. However, among the female breast cancer patients, survival rates were remarkably similar across younger and older age groups.Significant early mortality (within 6 months of diagnosis) was observed for liver, pancreatic, gallbladder, esophageal, stomach, lung, and brain cancers.

Overall survival in cancer patients is a comprehensive and unbiased measure of the entire disease trajectory. It reflects real-world outcomes and is easily understood by patients, healthcare providers, and researchers. As a patient-centric outcome, it directly correlates with patients’ well-being, reflecting the ultimate goal of cancer treatment—prolonging life [28]. By evaluating the duration from diagnosis to death due to any cause, OS offers a holistic perspective on the natural history and impact of the disease, considering all factors influencing a patient’s life expectancy, including disease progression, response to treatment, and potential complications. This approach is valuable for understanding the general prognosis of the cancer population and informing public health strategies. Additionally, OS analyses, together with net survival analyses, serve as a baseline for comparing the effectiveness of different treatments, guiding researchers, clinicians, and policymakers in developing and implementing strategies to improve outcomes across the entire spectrum of cancer care. While relative survival, or net survival, is a commonly applied measure, it solely reflects the impact of a cancer diagnosis on survival, ignoring other potential causes of death. In contrast, overall survival considers the likelihood of dying from cancer as well as from other competing causes, making it more pertinent for cancer patients and their treating clinicians [29]. Overall survival also offers benefits in specific scenarios, since it does not depend on the classification of the cause of death, which can vary significantly across different settings, time periods, countries, or populations being compared.

In line with international reports, we found much higher long-term (5-year) survival rates for certain cancer types, such as testicular, thyroid, breast, and prostate cancer, as well as melanoma, compared to more aggressive cancers, including stomach, lung, gallbladder, liver, and pancreatic cancer [1,30]. Differences in survival rates are driven by the stage of cancer at the time of diagnosis, its inherent aggressiveness and rate of growth, the efficacy of available treatment modalities, the presence of comorbidities, the overall health status and immune function of the patient, as well as genetic predispositions [31]. Furthermore, advancements in early detection techniques and the development of novel treatment approaches tailored to specific cancer types play a pivotal role in influencing survival outcomes [32]. Delayed diagnosis contributes to adverse outcomes in certain cancers, while early detection through screening programs improves prognosis in others. Delayed diagnosis can have significant implications for patient outcomes, particularly for cancers of the liver, lungs, pancreas, esophagus, and stomach, which are often diagnosed at advanced stages [33,34,35]. The effectiveness of advanced cancer treatments is significantly higher when applied in earlier stages of the disease, emphasizing the critical role of screening networks. However, the prolonged survival of patients undergoing these treatments has led to a continuous rise in oncological treatment costs, imposing an increasingly burdensome impact on society [10]. In our study, we found that approximately 50% of patients with these tumor types died within the first 6 months post-diagnosis. Several factors may contribute to delayed diagnosis, such as the lack of specific symptoms [36]. Lung, pancreatic, and stomach cancers often present with non-specific symptoms in the early stages, or the symptoms may be vague and easily mistaken for other common conditions [36,37,38]. This can lead to delays in seeking medical attention and subsequent diagnosis. Furthermore, the diagnosis of certain cancer types may be challenging due to the location of the tumors and the limitations of available diagnostic tests [33,34,35]. Overall, delayed diagnosis can result in more advanced disease at the time of diagnosis, which may limit treatment options and negatively impact patient outcomes. Raising awareness about the signs and symptoms of different cancers, improving access to timely diagnostic evaluation, and improving healthcare provider education can help mitigate delays in diagnosis and improve survival rates for affected individuals.

In high-income countries, cancers significantly contributing to the overall mortality are often those associated with high early mortality rates [19,39]. In 2019, a considerable proportion of diagnosed lung cancer patients died within 6 months of diagnosis, accounting for 2.8% of total deaths that year (3617 lung cancer deaths from 129,603 overall deaths in 2019) [40]. Similarly, significant numbers of individuals diagnosed with colorectal, pancreatic, stomach, and liver cancer died within the same timeframe in the same year.

While late-stage diagnosis is a key contributor to early mortality, our study did not directly assess tumor stage at diagnosis. Consequently, while delayed detection may partially explain high early mortality rates, additional contributing factors must be considered. These include the inherent aggressiveness of certain cancer types, inequities in accessing timely and effective treatment, and differences in healthcare system performance. Several studies have demonstrated that cancers with a particularly poor early survival—such as pancreatic, liver, and lung cancer—often exhibit rapid disease progression even when diagnosed at an early stage, diminishing the potential benefits of screening initiatives [1,30,41].

Moreover, differences in early mortality across countries can be influenced by broader systemic healthcare factors, such as treatment accessibility, diagnostic pathways, and post-diagnosis care quality. The CONCORD program and related global cancer survival studies have highlighted substantial international variability in early cancer survival, which cannot be solely attributed to differences in staging at diagnosis and also reflects disparities in healthcare system performance and treatment availability [30,41]. Future research integrating cancer registry data with clinical-stage information would be valuable for further clarifying the relationship between the stage at diagnosis and early mortality in Hungary.

Increasing the proportion of earlier-stage cancer diagnoses requires a multifaceted approach, addressing various aspects of healthcare delivery and public health initiatives. Implementing or improving the existing screening programs for common cancers can facilitate early detection. These programs should target high-risk populations and utilize effective screening modalities, such as colonoscopy for colorectal cancer and low-dose computed tomography (LDCT) for lung cancer in high-risk individuals.

In Hungary, after several pilot programs [25,42], organized colorectal screening was launched with the voluntary participation of general practitioners in the EFOP-1.8.1 priority project in late 2018. However, results from this initiative have not been published [43] or are very limited [44]. The Hungarian HUN-CHEST pilot program evaluated the feasibility of population-based lung cancer screening using low-dose CT screening in 1890 participants annually [45]. In the first round, 1.5% of participants were diagnosed with lung cancer, and most cases were detected at an early stage, indicating the potential of LDCT screening for early lung cancer diagnosis and treatment. However, nationwide lung cancer screening has not been implemented yet, although conceptual plans for a health economic analysis have been developed [46].

Despite the promise of screening in reducing cancer mortality, its impact on early mortality varies by cancer type. For example, colorectal and lung cancer screenings have been shown to improve early-stage detection and long-term survival [47,48]. However, for cancers with a very poor early survival, such as pancreatic or liver cancer, effective screening remains challenging due to the lack of highly sensitive and specific screening methods [49,50]. Furthermore, participation rates in screening programs also influence their effectiveness, and suboptimal adherence may reduce the expected benefits at the population level [51].

Unfortunately, well-established screening programs are not yet widely available for certain cancer types associated with significant early mortality, such as pancreatic, liver, gallbladder, and stomach cancer. For these cancers, early detection is often hindered by the lack of validated screening methods and the typically asymptomatic nature of early-stage disease. The rapid progression of these malignancies further limits the effectiveness of screening efforts, emphasizing the need for continued research into novel biomarkers and imaging techniques for early detection [52,53]. Additionally, disparities in healthcare access and diagnostic delays may disproportionately impact survival for these cancers, highlighting the importance of system-level interventions beyond screening alone [54].

In 2023, the Less Survivable Cancers Awareness Day was launched in the United Kingdom with the aim of highlighting the importance of cancers accounting for significant mortality that have historically received less research funding and public health attention [55]. Such initiatives underscore the need for enhanced research funding, public awareness campaigns, and targeted interventions to improve early detection and treatment access for cancers with high early mortality rates.

The HUN-CANCER EPI study revealed interesting changes in survival across various cancer types, with females showing a better survival probability compared to males. For example, during the 2015–2019 diagnostic period, women with lung or colorectal cancer had 4–8% higher survival rates compared to their male counterparts. The survival advantage of women has been extensively reported [56,57]. Sex-related disparities in cancer survival may be explained by tumor characteristics such as specific morphologies, as well as differences in risk factors such as hormone levels, infections, and chromosomal alterations [58,59]. Notably, the prevalence of smoking, a significant risk factor associated with higher cancer mortality rates, is substantially higher among males compared to females in the Hungarian adult population, which could impact cancer survival [60]. Moreover, previous research suggests a higher comorbidity burden among men at the time of diagnosis, which may also influence survival rates [61,62], and in the case of less aggressive tumors, the difference in general mortality between men and women also plays a significant role in this difference.

As part of the comprehensive assessment of cancer survival within the framework of the HUN-CANCER EPI study, we examined age-related differences in cancer survival across various age groups (0–18, 19–29, 30–39, 40–49, 50–59, 60–69, 70–79, and 80 years and above). For breast cancer and melanoma, younger individuals had similar survival probability compared to older cohorts. However, significant age-related disparities were observed among patients with lung, stomach, pancreatic, cervical, colorectal, kidney, ovarian, and pancreatic cancer. Several studies, including the SURVMARK study and a Slovenian survey, have reported significantly higher survival rates among younger individuals for cancers with poor prognosis, such as esophageal, stomach, pancreatic, and lung cancer. The difference was attributed to the better access of younger patients to adjuvant chemotherapy, and to their higher tolerance to more aggressive treatment regimens compared to older age groups. Advancements in diagnostic precision, staging technologies, and improved patient selection for targeted therapies based on molecular markers have also been noted as potential contributors to better survival among younger populations [5,63]. Age disparities in the overall survival appear to be less pronounced for cancers with higher survival rates and intensive screening programs, media coverage, and political campaigns (such as breast cancer, prostate cancer, and melanoma). In our study, breast cancer was associated with very similar long-term survival rates across most age cohorts, which may also be attributed to the higher incidence of certain more aggressive subtypes of BC in younger ages, such as triple-negative BC and HER2+ BC, which are associated with a worse prognosis [64,65].

The improvements observed in 5-year overall survival for almost all types of cancer between the 2011–2012 and 2017–2018 diagnostic periods are in line with the international trends reported over the past decade. Advances in cancer diagnostics, the introduction of more effective systemic therapies, and increased access to multimodal treatment approaches likely contributed to survival gains [32,66]. For example, the significant improvement in melanoma survival, particularly in males (HR = 0.57), may reflect the global impact of immune checkpoint inhibitors and targeted therapies, as shown in studies from Western Europe and North America [67,68]. Similarly, the survival improvements for colorectal cancer (HR = 0.76 in males, HR = 0.78 in females) mirror findings from other European countries where earlier detection through screening programs and improved surgical and adjuvant treatments have played pivotal roles [5,69]. For lung cancer, the modest yet significant improvement in males (HR = 0.89) is consistent with international findings, suggesting the growing contribution of targeted therapies and immunotherapy to better outcomes, despite the typically poor prognosis of this cancer type [30,70]. Breast cancer, with a hazard ratio of 0.68 in females between the two diagnostic periods, continues to reflect advances in early detection and personalized treatments, such as HER2-targeted therapies and hormonal interventions, a trend that is in line with outcomes reported in Scandinavian and Western European studies [30,71]. The steady improvement in prostate cancer survival (HR = 0.65) likely reflects advancements in androgen deprivation therapy and increased early-stage diagnoses through PSA screening, as observed globally [30,72]. These results suggest that Hungary’s cancer care improvements, while substantial, remain comparable to those achieved in higher-income countries, emphasizing the importance of ongoing investments in early detection programs, equitable access to modern therapies, and multidisciplinary treatment approaches. However, the magnitude of survival improvement for certain cancers indicates that further efforts are needed to reduce disparities and improve outcomes, particularly for cancers with poor prognosis.

When comparing survival outcomes internationally, it is important to acknowledge that population-based survival data can vary significantly between countries, which can influence the comparability of results. This is particularly true for cancers with high survival rates, where differences in demographic and healthcare factors across countries may complicate direct comparisons of OS outcomes. Moreover, OS data, which provide a clearer picture of cancer-specific survival by accounting for competing risks such as deaths from other causes, are less frequently reported in the literature than net survival (net-surv) results.

Despite these challenges, we believe that international comparisons remain essential for understanding the broader context of cancer survival outcomes. However, to address the limitations of OS data and provide a more nuanced perspective, we performed an independent net-surv analysis. This analysis, now published in a separate study, complements the OS findings and should be reviewed alongside them to gain a more comprehensive understanding of changes in survival [73].

In addition to advances in therapeutic procedures, early diagnosis, and screening, primary prevention plays a pivotal role in improving cancer survival outcomes. A healthy lifestyle, including a balanced diet, regular physical activity, and avoidance of smoking and excessive alcohol consumption, has been widely associated with a reduced risk of cancer and improved survival. Studies have demonstrated that regular physical activity can improve survival rates by enhancing immune function and reducing inflammation [74]. Additionally, proper nutrition, such as increased intake of fruits, vegetables, and whole grains, has been linked to a better prognosis in cancer survivors [75]. Obesity is a significant modifiable risk factor for several cancers, with weight control interventions improving both prevention and survival; however, studies suggest that greater adiposity at diagnosis does not necessarily worsen cancer survival, though variability in study designs and adiposity measurement methods complicates the interpretation of findings, highlighting the need for standardized approaches in future research [76]. Smoking cessation remains one of the most effective primary prevention strategies for cancer, as tobacco use is the leading cause of cancer mortality worldwide [77]. Public health campaigns promoting these preventive measures are essential in reducing cancer-related mortality and enhancing the long-term survival of patients.

The interpretation of our findings requires careful consideration of the strengths and limitations of the HUN-CANCER EPI study. A key strength lies in the substantial cohort size, with almost 530,000 cancer patients identified during the study period, increasing the statistical reliability of our findings. Rigorous data cleaning procedures were employed to ensure accuracy and validity. Furthermore, our methodology incorporated cancer-related interventions, allowing for the exclusion of cases where cancer-associated ICD codes were mistakenly assigned to a patient, thus enhancing the credibility of our conclusions. Additionally, the decade-long follow-up period provided a comprehensive picture of the changes in cancer survival over time. The nationwide coverage of the NHIF database enabled a thorough evaluation of cancer outcomes within the country.

Nevertheless, several limitations should be acknowledged. Our reliance on cancer-related ICD code records may have led to the exclusion of patients with secondary or multiple primary tumors, potentially resulting in an underestimated cancer incidence. The retrospective nature of our 11-year database analysis may not have captured cases where patients initially diagnosed with one primary tumor developed another type of primary cancer during the follow-up period, which influences the interpretation of results [78]. Moreover, our data lacked detailed information on histology, molecular pathology, staging, and Eastern Cooperative Oncology Group (ECOG) status, limiting our ability to assess survival rates by specific subtypes and examine the influence of patient-related factors on cancer survival.

The main value of this dataset lies in the fact that its results could serve as a fundamental resource for specialists involved in the treatment of cancer patients, shedding light on the outcomes of cancer management strategies for Hungarian patients over the past decade and providing essential insights for healthcare policy formulation and resource allocation. Furthermore, this comprehensive survival analysis offers crucial support and guidance to patients themselves, empowering them with valuable knowledge as they navigate the complexities of a cancer diagnosis. By elucidating the survival outcomes associated with different cancer types and demographic characteristics, our study significantly contributes to understanding the improvement of cancer care in Hungary, ultimately aiming to improve patient outcomes and quality of life. Another limitation of our study is the follow-up period, which ended on 30 September 2022. This cutoff prevented us from assessing 5-year survival outcomes for patients diagnosed between October 2017 and October 2019. Consequently, the survival analysis was incomplete for the most recent diagnostic periods. Future studies may benefit from extending the follow-up period to include these more recent diagnoses, enabling a more comprehensive assessment of long-term survival outcomes and providing a clearer picture of changes over time.

## 5. Conclusions

This study provides detailed survival data for cancer patients diagnosed in Hungary between 2011 and 2019, offering insights into the survival outcomes across various cancer types. Our findings reveal significant differences in survival rates by cancer type, with esophageal, pancreatic, liver, and lung cancers showing the poorest 5-year overall survival (OS) rates, ranging from 7.0% to 18.4%. In contrast, malignancies such as testicular cancer, thyroid cancer, Hodgkin’s lymphoma, and melanoma showed much higher survival rates, with 5-year OS exceeding 70%. Furthermore, the study highlights the significant improvements in survival rates for the patients diagnosed between 2017–2018 compared to those diagnosed in 2011–2012, indicating advancements in diagnostic techniques and treatment efficacy as well as developing primary prevention. Specifically, colorectal, lung, and melanoma cancers demonstrated notable survival improvements, with hazard ratios ranging from 0.57 to 0.89 depending on sex and cancer type. Additionally, our analysis showed that early mortality remains a major challenge, with a high proportion of patients dying within the first 2 to 6 months of diagnosis of such cancers as liver, lung, pancreatic and esophageal cancers. These findings underscore the importance of early detection and timely interventions to improve survival outcomes, particularly for high-mortality cancers. The study also highlighted significant sex-related disparities in cancer survival, with females generally exhibiting better survival rates than males across most cancer types. These results emphasize the need for continued efforts in improving early diagnosis, targeted treatments, and personalized care to enhance cancer survival and reduce early mortality.

## Figures and Tables

**Figure 1 cancers-17-01670-f001:**
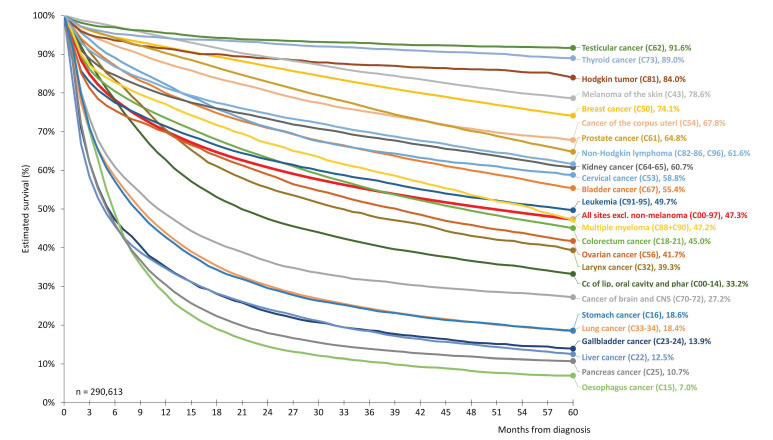
Estimated long-term survival of Hungarian cancer patients by different cancer types according to the International Statistical Classification of Diseases and Related Health Problems, 10th Revision (diagnostic period: 2015–2019). CNS: central nervous system. All results reflect C00–97, excluding C44 and C77–80.

**Figure 2 cancers-17-01670-f002:**
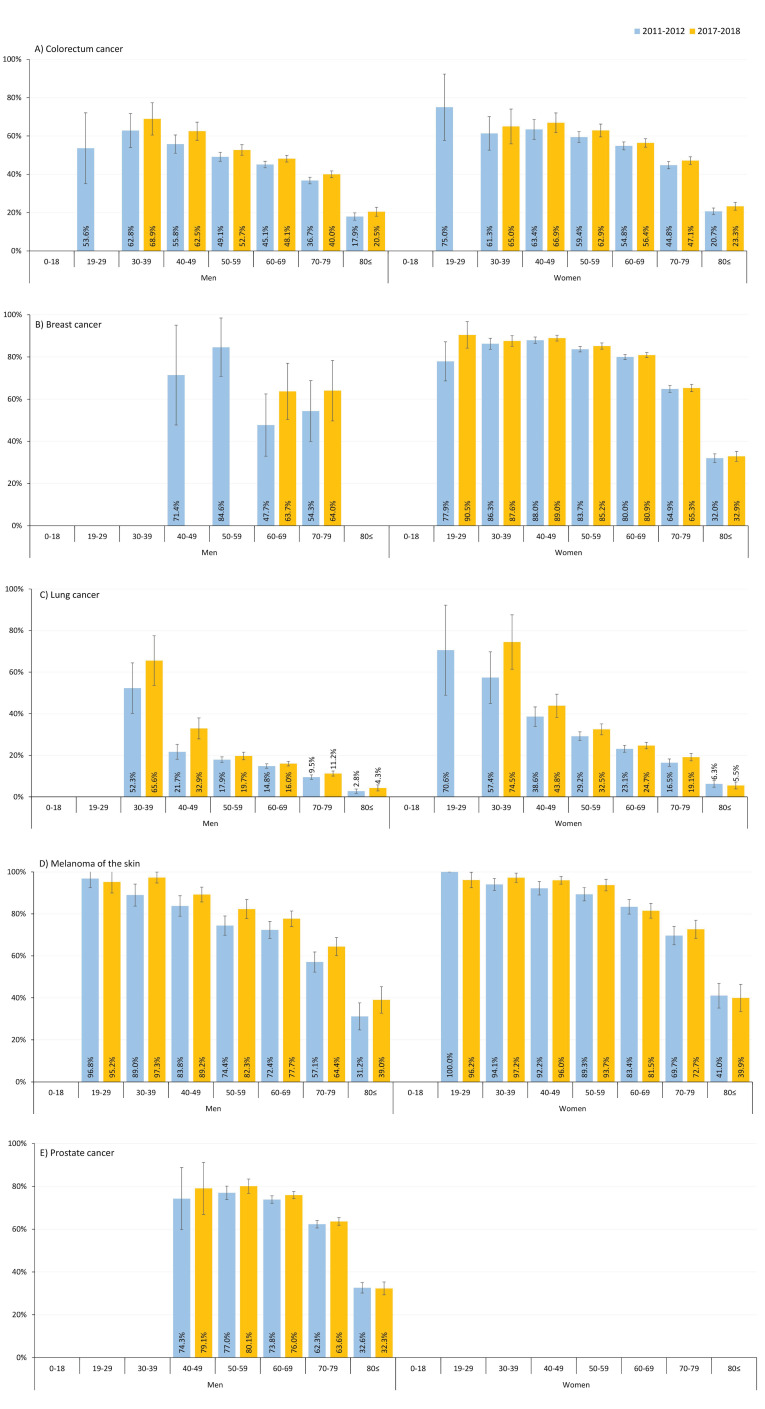
Estimated 5-year overall survival by age and sex for the most common cancer types among the patients diagnosed between 2011–2012 and 2017–2018. (**A**) Colorectum cancer; (**B**) Breast cancer; (**C**) Lung cancer; (**D**) Melanoma of the skin; (**E**) Prostate cancer.

**Figure 3 cancers-17-01670-f003:**
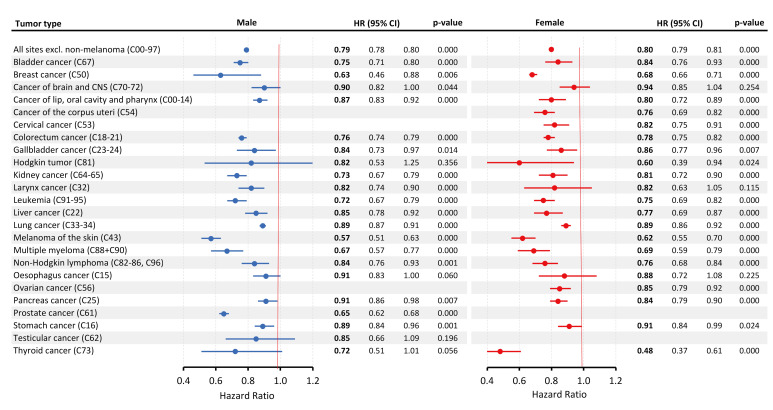
Changes in mortality rates between diagnostic periods by sex (2017–2018 vs. 2011–2012), expressed as hazard ratios. HR: hazard ratio, CI: confidence interval.

**Figure 4 cancers-17-01670-f004:**
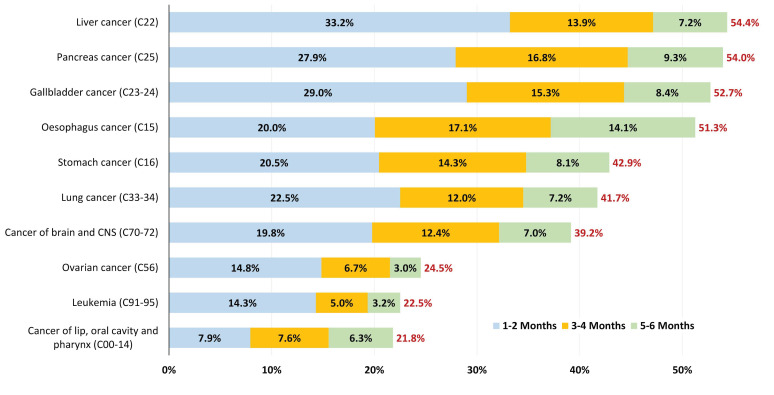
Percentages of the patients diagnosed in 2015–2019 who died within the first 2, 4, and 6 months after cancer diagnosis for the top 10 most fatal tumor types. CNS: central nervous system.

**Figure 5 cancers-17-01670-f005:**
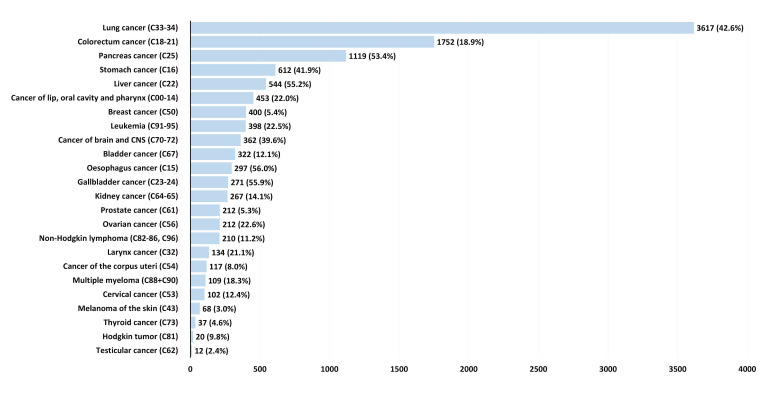
Number (and percentages) of deaths within the first 6 months after diagnosis for the patients diagnosed with cancer in 2019. CNS: central nervous system.

## Data Availability

Data sharing is available in Appendix A, and the authors are happy to share all available data from the study.

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
