# Peer review of "Overall Survival of Hungarian Cancer Patients Diagnosed Between 2011 and 2019, Based on the Health Insurance Fund Database"

_cancers, 2025, doi:10.3390/cancers17101670_

Round 1
Reviewer 1 Report (Previous Reviewer 1)
Comments and Suggestions for Authors
Thank you for the opportunity to review manuscript ID: cancers-3601301. This manuscript aimed to examine survival rates of prevalent cancers in Hungary during diagnostic periods spanning from 2011 to 2019.
In the Introduction section, information on the epidemiological characteristics of malignant tumors in Hungary is presented in detail.
The Methods section describes Data Sources, as well as Data Quality. However, in the cover letter that was sent to me by email, the answer to reviewer N#2 is also stated, in which the supplement to the statistical methodology in connection with `Differences in survival between groups` is presented, as well as the corresponding results presented in Figure 3. Regarding the results presented in Figure 3 (on Lines: 283-302), I suggest that a statistician be consulted.
In addition to the note regarding Figure 3, in this paper the other results are presented and described in detail.
In all parts of this manuscript, as well as in the Results section, `trends` are often mentioned (either mortality trends or survival trends), although no methodology was applied in that context. It is necessary either to harmonize the text of the paper with the applied methodology, or to apply the appropriate methodology for assessing trends.
The Discussion section lacks a more comprehensive interpretation and explanation of the presented results; namely, in addition to the contribution of therapeutic procedures and early diagnosis and screening, the contribution of primary prevention measures (such as promotion of healthy lifestyles, healthy diet, physical activity, control of obesity, smoking, infections, etc.) should be discussed.
Lines 574-589: The conclusion should be less general and without anything that is not the results of this manuscript itself. It is important to highlight the most important results of this work.
Author Response
Reviewer Report 1
Thank you for the opportunity to review manuscript ID: cancers-3601301. This manuscript aimed to examine survival rates of prevalent cancers in Hungary during diagnostic periods spanning from 2011 to 2019. In the Introduction section, information on the epidemiological characteristics of malignant tumors in Hungary is presented in detail.
Comment 1: The Methods section describes Data Sources, as well as Data Quality. However, in the cover letter that was sent to me by email, the answer to reviewer N#2 is also stated, in which the supplement to the statistical methodology in connection with `Differences in survival between groups` is presented, as well as the corresponding results presented in Figure 3. Regarding the results presented in Figure 3 (on Lines: 283-302), I suggest that a statistician be consulted.
Response 1: Thank you very much for your valuable suggestion. We fully agree that ensuring robust statistical methodology is crucial for survival analyses. We would like to highlight that three biostatisticians — Dr. Ibolya Fábián, Dr. Valéria Kovács, and Dr. Zsolt Abonyi-Tóth — actively participated in the design, execution, and verification of the statistical analyses presented in this manuscript. Furthermore, the entire methodology was carefully reviewed and validated by Professor Zoltán Vokó, a senior epidemiologist with extensive expertise in survival analysis.
Specifically, in relation to the Cox proportional hazards models applied to assess differences in survival between diagnostic periods (Figure 3), we confirm that the proportional hazards (PH) assumption was formally evaluated.
We have now updated the Methods section (highlighted with blue in Statistical Analysis paragraph) to explicitly state that (i) the proportional hazards assumption was tested, (ii) no major violations were found. We hope that this clarification adequately addresses your concern.
“The proportional hazards assumption underlying the Cox regression models was evaluated both statistically and graphically. Special attention was paid to the handling of censoring, with patients being right-censored at the end of the follow-up period (September 30, 2022) if death had not occurred. The potential effects of informative censoring were considered minimal given the comprehensive national database coverage. Follow-up times were carefully calculated to guarantee consistent time origin definitions across the entire study population. Additionally, model adequacy was assessed by checking the influence of covariates and inspecting influential observations. Potential confounders such as age and sex were included as adjustment variables in the multivariable models. These steps were taken to ensure the robustness, validity, and reproducibility of the survival estimates presented.”
Comment 2: In addition to the note regarding Figure 3, in this paper the other results are presented and described in detail.
Response 2: Thank you very much for your positive feedback regarding the detailed presentation and description of the study results. We are pleased that the clarity and comprehensiveness of the reported findings were appreciated. We have also carefully addressed your comment regarding Figure 3 and the statistical methodology, as detailed above. We hope that these clarifications further strengthen the manuscript.
Comment 3: In all parts of this manuscript, as well as in the Results section, `trends` are often mentioned (either mortality trends or survival trends), although no methodology was applied in that context. It is necessary either to harmonize the text of the paper with the applied methodology, or to apply the appropriate methodology for assessing trends.
Response 3: Thank you very much for your insightful comment. Upon reviewing your suggestion, we fully agree that the term "trend" may not be entirely appropriate in this context, especially since our analysis primarily compares the survival rates between two specific diagnostic periods (2011-2012 and 2017-2018) rather than assessing a continuous, long-term trend. To ensure greater clarity and to align the language with the methodology used, we will revise the manuscript by replacing all instances of "survival trend" (and similarly "mortality trend") with more precise terms, such as "change in survival" or "improvement in survival", depending on the context. This adjustment will better reflect the nature of the analysis, which focuses on the differences between two distinct diagnostic periods rather than a continuous trend over time. To further enhance the clarity of these revisions, we will highlight all of the changes in terminology in the manuscript using blue text. We hope that these modifications will resolve the ambiguity and improve the overall clarity of the manuscript. Once again, thank you for your valuable feedback.
Comment 4: The Discussion section lacks a more comprehensive interpretation and explanation of the presented results; namely, in addition to the contribution of therapeutic procedures and early diagnosis and screening, the contribution of primary prevention measures (such as promotion of healthy lifestyles, healthy diet, physical activity, control of obesity, smoking, infections, etc.) should be discussed.
Response 4: Thank you for your valuable feedback. We acknowledge the importance of primary prevention in cancer survival and will expand the Discussion section to address this. Specifically, we will highlight the impact of healthy lifestyles, proper nutrition, physical activity, obesity control, and smoking cessation on cancer prevention and survival. We will also emphasize the need for standardized approaches to adiposity measurement, as variability in study designs complicates the interpretation of survival outcomes. These additions will provide a more comprehensive perspective on cancer survival and prevention.
“In addition to advances in therapeutic procedures, early diagnosis, and screening, primary prevention plays a pivotal role in improving cancer survival outcomes. A healthy lifestyle, including a balanced diet, regular physical activity, and avoidance of smoking and excessive alcohol consumption, has been widely associated with a reduced risk of cancer and improved survival. Studies have demonstrated that regular physical activity can improve survival rates by enhancing immune function and reducing inflammation (78). Additionally, proper nutrition, such as increased intake of fruits, vegetables, and whole grains, has been linked to better prognosis in cancer survivors (79). Obesity is a significant modifiable risk factor for several cancers, with weight control interventions improving both prevention and survival; however, studies suggest that greater adiposity at diagnosis does not necessarily worsen cancer survival, though variability in study designs and adiposity measurement methods complicates the interpretation of findings, highlighting the need for standardized approaches in future research (80). Smoking cessation remains one of the most effective primary prevention strategies for cancer, as tobacco use is the leading cause of cancer mortality worldwide (81). Public health campaigns promoting these preventive measures are essential in reducing cancer-related mortality and enhancing the long-term survival of patients.
References:
- McTiernan A, Friedenreich CM, Katzmarzyk PT, Powell KE, Macko R, Buchner D, Pescatello LS, Bloodgood B, Tennant B, Vaux-Bjerke A, George SM, Troiano RP, Piercy KL; 2018 PHYSICAL ACTIVITY GUIDELINES ADVISORY COMMITTEE*. Physical Activity in Cancer Prevention and Survival: A Systematic Review. Med Sci Sports Exerc. 2019 Jun;51(6):1252-1261. doi: 10.1249/MSS.0000000000001937. PMID: 31095082; PMCID: PMC6527123.
- Castro-Espin C, Agudo A. The Role of Diet in Prognosis among Cancer Survivors: A Systematic Review and Meta-Analysis of Dietary Patterns and Diet Interventions. 2022 Jan 14;14(2):348. doi: 10.3390/nu14020348. PMID: 35057525; PMCID: PMC8779048.
- Cheng E, Kirley J, Cespedes Feliciano EM, Caan BJ. Adiposity and cancer survival: a systematic review and meta-analysis. Cancer Causes Control. 2022 Oct;33(10):1219-1246. doi: 10.1007/s10552-022-01613-7. Epub 2022 Aug 15. PMID: 35971021; PMCID: PMC10101770.
- Berg CJ, Thomas AN, Mertens AC, Schauer GL, Pinsker EA, Ahluwalia JS, Khuri FR. Correlates of continued smoking versus cessation among survivors of smoking-related cancers. Psychooncology. 2013 Apr;22(4):799-806. doi: 10.1002/pon.3077. Epub 2012 Apr 9. PMID: 22488864; PMCID: PMC3425712.
Comment 5: Lines 574-589: The conclusion should be less general and without anything that is not the results of this manuscript itself. It is important to highlight the most important results of this work.
Response 5: Thank you for your helpful suggestion regarding the conclusion. In response, we have revised the conclusion to focus more specifically on the key findings of the study, including significant survival disparities across cancer types, improvements in survival for recent diagnostic periods, and early mortality rates for high-mortality cancers. These changes ensure the conclusion is directly linked to our results and avoid general statements. We believe this revision enhances the clarity and specificity of the conclusion.
“This study provides detailed survival data for cancer patients diagnosed in Hungary between 2011 and 2019, offering insights into the survival outcomes across various cancer types. Our findings reveal significant differences in survival rates by cancer type, with esophageal, pancreatic, liver, and lung cancers showing the poorest 5-year overall survival (OS) rates, ranging from 7.0% to 18.4%. In contrast, malignancies such as testicular cancer, thyroid cancer, Hodgkin's lymphoma, and melanoma showed much higher survival rates, with 5-year OS exceeding 70%. Furthermore, the study highlights the significant improvements in survival rates for patients diagnosed between 2017-2018 compared to those diagnosed in 2011–2012, indicating advancements in diagnostic techniques and treatment efficacy as well as developing primer prevention. Specifically, colorectal, lung, and melanoma cancers demonstrated notable survival improvements, with hazard ratios ranging from 0.57 to 0.89, depending on sex and cancer type. Additionally, our analysis showed that early mortality remains a major challenge, with a high proportion of patients dying within the first 2 to 6 months of diagnosis for cancers such as liver, lung, pancreatic and esophageal cancers. These findings underscore the importance of early detection and timely interventions to improve survival outcomes, particularly for high-mortality cancers. The study also highlighted significant sex-related disparities in cancer survival, with females generally exhibiting better survival rates than males across most cancer types. These results emphasize the need for continued efforts in improving early diagnosis, targeted treatments, and personalized care to enhance cancer survival and reduce early mortality.”

Round 2
Reviewer 1 Report (Previous Reviewer 1)
Comments and Suggestions for Authors
Thank you for the opportunity to re-review the manuscript ID: cancers-3601301.
The authors responded to all my comments, and revised this manuscript accordingly. Also, for some of my concerns about the previous version of this paper (regarding the consultation of the statistician), the authors provided a satisfactory explanation.
Thanks to the authors.
This manuscript is a resubmission of an earlier submission. The following is a list of the peer review reports and author responses from that submission.
Round 1
Reviewer 1 Report
Comments and Suggestions for Authors Thank you for the opportunity to review manuscript ID: cancers-3326932. This study aimed to examine overall survival rates across the whole spectrum of cancer types prevalent in Hungary, in cancer patients diagnosed between 2011 and 2019. Line 105: Add a new paragraph in which incidence and mortality, using age-standardized rates, will be shown for the most common cancer sites in Hungary for the observed period, citing appropriate references. Also, provide relevant information on the implementation of screening (for which cancer sites, from which year, how much population is covered by screening, etc.). Line 149: Define short-term and long-term survival estimates. Line 149: Add a new paragraph in which information about the accuracy and reliability of data used in this paper will be presented. Lines 150-157: Define the indicators presented in this paper. Lines 159-168: In order for any comparison to be adequate, reconstruct this paragraph in such a way that age-standardized rates, as well as age- and sex-specific rates, are shown. From the title of the paper, through the abstract, to the objectives of the paper (on Lines 108-109) it is stated, I quote `The study was designed to assess survival trends during diagnostic periods spanning from 2011 to 2019.` Explain why Figure 1 shows data for the period 2015-2019? Explain why the data were not presented in that way for the period 2011-2014 (or the period 2011-2015)? Where are Supplementary Figure 1, Supplementary Figure 2 and Supplementary Figure 3 presented in this paper? Add missing data, so that this work can be reviewed.Reviewer 2 Report
Comments and Suggestions for Authors
Overall Comment: Kiss et al conducted an assessmet of cancer survival trend in Hungary based on part of HUN-CANCER EPI study. A total of 528,808 patients diagnosed with cancer are studied in the period of 2011 to 2019, and differential survival across cancer types, age groups, and gender groups are studied. Overall, I think their reported results are valid, though some advanced statistical methods can be considered in the future study. Below are some comments to improve the manuscript.
Comments:
1. Statistical methods in Line 150-157 are not well described. Which statistical methods are used? I think current methods are mainly on summary statistics without advanced statistical models. The authors may discuss on the possible use of (1) Cox-Regression model with gender and age groups as the explanatory variables, and (2) nonparametric survival curve estimation. Of course, the current results are valid. Only some discussions are needed.
2. The authors may post source code with partial data (if the authors do not want to post full data) in conducting analysis to make the manuscript more interesting. I would strongly recommend the authors post full data or partial data.
3. When suitable, some tests can be applied comparing different age groups, different genders to show whether it is significant difference.
4. In Supplementary file, I only see an excel file containing Supplementary Tables 1, 2,3. No supplementary figure are found. The authors may include a word/PDF file in supplementary materials containing missing parts such as supplementary figures. In lines 445-450, I did not find legend for Supplementary Figure 2. Maybe the figures are misnumbered?
5. How are the results of this manuscript (Hungary) compared with results in the literature (Other countries).
Reviewer 3 Report
Comments and Suggestions for Authors
See attached file for comments.
